# The Outline of Seed Silhouettes: A Morphological Approach to *Silene* (Caryophyllaceae)

**DOI:** 10.3390/plants11233383

**Published:** 2022-12-05

**Authors:** José Javier Martín-Gómez, José Luis Rodríguez-Lorenzo, Ángel Tocino, Bohuslav Janoušek, Ana Juan, Emilio Cervantes

**Affiliations:** 1IRNASA-CSIC, Cordel de Merinas, 40, 37008 Salamanca, Spain; 2Plant Developmental Genetics, Institute of Biophysics v.v.i, Academy of Sciences of the Czech Republic, Královopolská 135, 61265 Brno, Czech Republic; 3Departamento de Matemáticas, Facultad de Ciencias, Universidad de Salamanca, Plaza de la Merced 1-4, 37008 Salamanca, Spain; 4Departamento de Ciencias Ambientales y Recursos Naturales, University of Alicante, San Vicente del Raspeig, 03690 Alicante, Spain

**Keywords:** Caryophyllaceae, circularity, convexity, echinate seeds, roundness, rugose seeds, smooth seeds, solidity of seeds

## Abstract

Seed morphology is an important source of information for plant taxonomy. Nevertheless, the characters under study are diverse, and a simple, unified method is lacking in the literature. A new method for the classification of seeds of the genus *Silene* based on optical images and image analysis has recently been described on the basis of morphological measurements of the lateral seed views. According to the outline of their silhouettes, seeds from 52 species (49 of *Silene* and three related species) were classified in three groups: smooth, rugose and echinate, revealing remarkable differences between these groups. This methodology has been applied here to 51 new species, making a total of 100 species of *Silene* analyzed so far. According to our data, a new group was described, termed papillose. The results showed morphological differences between the four mentioned seed groups, with reduced values of circularity for dorsal and lateral seed views in the papillose and echinate groups and reduced values of solidity in the papillose seeds. The method was applied to the analysis of individual as well as to average seed silhouettes and some of the differences between groups were maintained in both cases.

## 1. Introduction

The genus *Silene* L. with ca. 700 species is the largest one of the Caryophyllaceae Juss. [1,2]. An important proportion of the *Silene* species belong to the Northern Hemisphere, with regions of high diversity in the Mediterranean area [3,4,5,6,7,8,9,10], Iran and western Asia [11,12,13]. Other species have been described from India and the Himalayas [14], and even from China [15]. In addition, a scarce number of species grow in South Africa (e.g., *S. burchellii* Otth. [16] and *S. undulata* Aiton [17]), or in diverse locations along South America, as the montane tropical biomes of Ecuador, Peru, Bolivia and Argentina (e.g., *S. thysanodes* Fenzl. and *S. mandonii* (Rohrb.) Bocquet) [18,19].

The diversity in shape and ornamentation observed in the seeds of *Silene* has been recognized for a longtime as a valuable source of information for intrageneric classification, and seed morphological characters have been used with diverse applications for the taxonomy of this genus [20,21,22,23,24,25,26,27,28]. There is a general agreement concerning the characters that may be used, including overall seed shape, and diverse aspects of seed coat ornamentation. Nevertheless, the conformity is not complete about how these characters should be defined, and the level of precision that should be reached in their description.

The shape of lateral seed views, with the hilum oriented towards one side, has often been termed as reniform, and it could be more precisely defined as “cardioid related”, based on the quantification by comparison with models derived from the cardioid curve. A variable degree of opening of the central side, corresponding to the proximity of the hilum, revealed different shapes that can be compared with algebraically defined geometric models derived from the cardioid [7,29,30,31,32]. In their dorsal views, the seeds may be convex or show concavities. Convexity can be quantified in geometry by the measurement of solidity. Solidity is the relationship between the area of a plane image and the area of the corresponding convex Hull [33]. In convex figures, solidity equals 1. The seeds with large concavities in their dorsal views correspond to those that were termed dorso-canaliculata by Boissier and Rorbach [20,21], and thus solidity of the dorsal views of seeds might be an important property for the taxonomical characterization of *Silene* species.

Concerning seed coat ornamentation, there is not a complete agreement about the terminology and definition of the characters under study. For example, Arman and Jolipour [11] classified the seeds by their surface structure in papillate, tuberculate and smoothness, while Hoseini et al. [25] classify the cell surface in four categories: flat, convex, mammalian, echinate, and tuberculate. Atazadeh et al. [26] analyzed the character testa cell’s edge, classifying it as v-shaped, undulate, smooth, and sinuate, while Ullah et al. [27] defined four different groups: wavy, irregular, tetragonal, and elongated, based on the character epidermal cells structure. Although for certain features there is a conformity of the characters analyzed (e.g., cell or seed surface), a difficulty derives from possible variations in their nomenclature and interpretation.

The proposed description of the lateral and dorsal views of *Silene* seeds was based on the consideration of the seed silhouettes as geometric figures. In relation to general seed shape, it allowed for comparing the seed outline with algebraically defined geometric figures providing a quantity, *J* index, that indicates the percentage of similarity of the seed silhouette with a given geometric figure taken as a model [7,29,30,31,32]. Related to seed coat ornamentation, and working specifically with seed silhouettes, we have analyzed recently 100 populations belonging to 49 species of *Silene*, one of *Atocion* and two of *Viscaria* [31]. The seeds were classified according to the outline of their silhouettes in three groups described as smooth, rugose or echinate [31]. These three morphological groups were related also to the subgeneric classification based on DNA sequence analysis [34]. The group of smooth seeds contained mainly species corresponding to *S.* subg. *Silene*, while species of *S.* subg. *Lychnis* were absent. Most of the species in the group of rugose seeds belonged to *S.* subg. *Silene*, and species in the group of echinate seeds corresponded mainly to both *S.* subg. *Behenantha* and *S.* subg. *Lychnis* [31].

In addition, previous published data indicated a relationship between the morphological groups defined based on their seed silhouettes (i.e., smooth, rugose and echinate) and the geometric models adjusting to the shape of seeds, both in the lateral and dorsal views [31,32]. Lateral models (LM) LM3, LM5 and LM6 were characterized by an open (concave) region around the hilum and were predominant in the groups of smooth and rugose seeds, while the models LM2 and LM4, more closed around the hilum region, were more frequent in the group of echinate seeds. Regarding the dorsal models (DM), the convex figures termed DM1 to DM4 adjusted better to seed shape of echinate seeds, while figures DM5 to DM9, having diverse concavities, adjusted to seed shape of smooth seeds [32].

To explore further the relationship between morphological groups based on the geometry of the seed silhouettes and the taxonomic position of the species, we have expanded the seed morphological analysis to 51 new species of *Silene*, making a total of 100 species of *Silene* analyzed so far by this method. The species *Eudianthe coeli-rosa* (L.) Fenzl ex Endl. was included since this taxon has also been considered as a member of the genus *Silene* [2]. Our aim was to apply the seed classification method previously described [31], based on the seed silhouettes obtained from the analysis of photographic images together with their morphological seed measures and hence to test its utility as a source of information applicable for morphological and taxonomic purposes.

## 2. Results

### 2.1. General Morphological Analysis

The lateral and dorsal images of representative seeds belonging to 51 species of *Silene* and *Eudianthe coeli-rosa* used in this study are shown in Appendix B (Figure A1). The seed images used in the analysis are stored in Zenodo (see Appendix A). Table 1 contains a summary of the results concerning the general morphological analysis of the studied *Silene* seeds with the mean, minimum and maximum values of area, perimeter, circularity, roundness, aspect ratio, and solidity. Similar to other *Silene* species [31,32], the coefficient of variation was higher for the area and perimeter values. The minimum values of this coefficient were obtained for the measurements of solidity, whose ranges were clearly close.

### 2.2. Morphological Types Based on the Outlines of the Seed Silhouettes

Among the studied *Silene* species, only four (*S. baccifera*, *S. damascena*, *S. legionensis* and *S. pomelii* subsp. *adusta*) had a smooth outline on the lateral view, since no projections appeared on their surface (Figure 1). The seeds of three of these species presented partial concavities in their lateral and dorsal seed views (Figure 1 and Figure 2), in contrast to the dorsal seed views of *S. baccifera*, which were clearly convex. Most of these species belong to *S*. subg. *Silene* (3 out of 4, 75%), with a unique sample of *S*. subg. *Behenantha* (*S. baccifera*).

The group of rugose seeds was the largest in number, formed by 27 species (Figure 3): *S. bupleuroides*, *S. caryophylloides*, *S. chloranta*, *S. chlorifolia*, *S. dinarica*, *S. foliosa*, *S. frivaldskyana*, *S. gigantea*, *S. hayekiana*, *S. hookeri*, *S. integripetala*, *S. koreana*, *S. linicola*, *S. marizii*, *S. multicaulis* subsp. *multicaulis*, *S. paradoxa*, *S. pygmaea*, *S. regia*, *S. roemeri*, *S. saxatilis*, *S. spinescens*, *S. squamigera* subsp. *vesiculifera*, *S. swertifolia*, *S. vallesia* subsp. *vallesia*, *S. villosa*, *S. virginica* and *S. waldsteinii*. As with smooth seeds, most of the species with rugose seeds correspond to *S*. subg. *Silene* (22 out of 27, 81.5%), while only five species belong to *S*. subg. *Behenantha*. A notable diversity about the degree of concavity around the hilum was observed, with species characterized with this region almost plane or convex (e.g., *S. marizii*, *S. virginica*) or with some degree of concavity (e.g., *S. caryophylloides*, *S. linicola*). Regarding the dorsal seed view (Figure 4), most of the studied species showed a rectangle shape with slightly to moderate depression in the upper and lower sides of the seeds (e.g., *S. bupleroides*, *S. gigantea*, *S. roemeri*, *S. villosa*), while a few species were characterized with convex dorsal views without any depression (e.g., *S. dinarica*, *S. koreana*).

The group of echinate seeds is formed by the following 16 species (Figure 5): *S. aprica*, *S. chungtienensis*, *S. ciliata*, *S. dichotoma*, *S. fabaria*, *S. firma*, *S. fruticosa*, *S. jeniseensis*, *S. longicilia*, *S. multiflora*, *S. nana*, *S. petersonii*, *S. samojedorum*, *S. suksdorfii*, *S. viridiflora* and *S. yunannensis*. Most of the species in this group belong to *S.* subg. *Behenantha* (10 species out of 16, 62.5%), while only six species belong to *S. subg.*
*Silene*. In general, the lateral views of the seeds are plane around the hilum region (e.g., *S. fruticosa*) or present concavities smaller than the rugose seeds (e.g., *S. aprica*, *S. dichotoma*). A notable diversity in size and relative distribution of the spines was observed for both lateral and dorsal seed views (Figure 5 and Figure 6). Conversely to smooth and rugose seeds, the dorsal seed views of the echinate seeds were mostly convex (excluding the concavities due to the spines), or rarely truncate rectangular (e.g., *S. fruticosa* and *S. longicilia*, respectively), with a clear concavity in the upper or lower part of the seeds only observed in *S. ciliata* (Figure 6).

In addition, some species did not match with the previous groups, and another new category was defined as papillose (Figure 7 and Figure 8). The papillose seeds are characterized at their cell surface by longer and broader prolongations (papillae) than the spines of the echinate group. In addition, although some papillae can be acute, most of them are flat at their tips. Four species were included in this group of papillose seeds (Figure 7): *S. holzmannii*, *S. laciniata*, *S. magellanica* and *S. perlmanii*. Most of these species belong to *S*. subg. *Behenantha* (3 out of 4, 75%). Their dorsal silhouettes showed a rectangular shape, with the presence of marked prolongations (Figure 8).

### 2.3. The Average Silhouettes in Each Morphological Group

When compared to the individual silhouettes, the average silhouettes give a representation of the overall seed shape in the group of 20 seeds representing one species. The average silhouettes obtained for each group are shown in Appendix C (Figure A2, Figure A3, Figure A4 and Figure A5), and the results were rather coincident with those exposed for individual seed outline silhouettes in each morphological group. In the group of smooth seeds (Figure A2), the lateral views presented a notable concavity in the hilum region for *S. damascena, S. legionensis* and *S. pomeli* subsp. *adusta* but not in *S. baccifera*. In the dorsal views, the seeds of *S. legionensis* and *S. baccifera* were convex, while those of *S. damascena* and *S. pomeli* subsp. *adusta* present concavities in their upper and lower sides.

The comparison between the average silhouettes of the rugose with the echinate seeds (Figure A3 and Figure A4) shows that the lateral views of the seeds were more closed in the region around the hilum in the echinate than the rugose seeds. Moreover, the process of obtention of the average seed silhouette resulted in a softened outline surface, with less apparent rugosities in both seed groups. Nevertheless, the average silhouettes of the echinate seeds maintained a slightly more uneven outline than the rugose seeds. This effect is still more remarkable in the average silhouettes of the papillose seeds, that still maintained a more uneven outline (Figure A5).

### 2.4. Morphological Comparison between Seed Groups

The mean values for all general morphological measurements, coefficients of variation and the results of the Kruskal–Wallis and post hoc tests for the comparison between the groups of *Silene* seeds (smooth, rugose, echinate, and papillose), as well as *E. coeli-rosa* are shown in Table 2 and Table 3, for the individual outline silhouettes of the lateral and dorsal seed views respectively.

On the one hand, Kruskal–Wallis test showed differences (*p* < 0.05) on the lateral views between the groups for all the measurements except aspect ratio and roundness. These measurements were excluded from further analysis. For the remaining measures, the Campbell and Skillings post-hoc test results are summarized in Table 2. It can be concluded that echinate and *E. coeli-rosa* populations have lower values for area and perimeter, whereas the papillose seeds have a high perimeter (*p* < 0.05). Regarding the circularity and solidity, there were four groups, with the lowest values in the papillose seeds, followed by echinate and *E. coeli-rosa*, then rugose and the highest values of the smooth seeds (*p* < 0.05). The obtained data for the seeds of *E. coeli-rosa* were, in general, similar to those for the echinate seeds in all measurements. The coefficients of variation for the lateral seed view were notably higher for the measurements of area and perimeter, intermediate in aspect ratio, circularity and roundness, and the lowest values were obtained in solidity (Table 2).

On the other hand, related to the dorsal views, there were differences between the groups for all the measurements, with values of area and perimeter lower in *E. coeli-rosa* and the echinate seeds than in the other groups (*p* < 0.05; Table 3), and the highest values of perimeter in the papillose group. The seeds were classified by roundness in three groups: (i) the group of lowest roundness included both the smooth and rugose seeds; (ii) the second group corresponded to the echinate seeds with intermediate values of roundness (*p* < 0.05); and, finally, (iii) a group characterized by the highest values of roundness (*p* < 0.05) formed by the papillose and *E. coeli-rosa* seeds. The circularity results show lowest values for the papillose population, followed by echinate, and, finally, *E. coeli-rosa*, the rugose and smooth species with the highest values (*p* < 0.05). The lowest values of solidity were obtained for the papillose species, an intermediate group was formed by echinate and smooth seeds, and the highest values corresponded to the rugose population. The species *E. coeli-rosa* resembled different seed groups depending on the morphological feature: echinate seeds for area, perimeter and solidity; papillose seeds for aspect ratio and roundness; and rugose and smooth seeds for solidity and circularity. Finally, the coefficients of variation for the dorsal seed view revealed the highest values for the area and perimeter, the lowest values were obtained in solidity, while aspect ratio, circularity and roundness showed similar intermediate values (Table 3).

A comparison between groups was made also with the data obtained from the average silhouettes for each species. No differences were found in the measurements of aspect ratio and roundness in the lateral views and for all the measurements in the dorsal views. Table 4 contains the comparative results of the circularity and solidity of average silhouettes between the four groups in the lateral seed views. The comparison revealed no differences between the three main groups (smooth, rugose, echinate). Differences were found between papillose and the other groups in circularity, and between papillose and echinate and rugose in solidity.

From the comparison of the data based on the individual silhouettes (Table 2) and the average silhouettes (Table 4), the coefficients of variation were notably reduced for the data obtained on the later, since great part of the variation was lost during the process of obtention of the average silhouette. Nevertheless, this process resulted in a loss of variation higher for the echinate seeds and lower for the smooth seeds, for circularity, roundness and solidity values of the lateral views,

### 2.5. The Relationship between Groups and Taxonomic Sections

The results reported here confirm and expand a certain relationship between the morphological groups obtained by the analysis of the outline of the silhouettes. A summary of them is presented in Table A1 in the Appendix D. Most species in the smooth and rugose groups belong to *S.* subgen. *Silene*, while those in the echinate group belong predominantly to *S.* subgen. *Behenantha* and *S.* subgen. *Lychnis*. Among those species from *S.* subgen. *Silene* adscribed to the group of echinate seeds, a majority (7 out of 8) belong to S. Sect. *Siphonomorpha*. In the cases of *S.* subgen. *Behenantha,* 3 out of 8 species of the group of rugose seeds are included in *S.* sect. *Physolychnis*. Finally, the case of *S. baccifera* and *S. littorea*, of *S.* subgen. *Behenantha* are quite remarkable, since both of them are characterized by a convex shape and bright surface, and hence, they are the only species belonging to this subgenus and classified within the smooth group.

## 3. Discussion

A recent report with 95 populations belonging to 52 species (49 species of *Silene* and 3 related species) classified the seeds according to the outline of their silhouettes in 3 well distinguished groups: smooth, rugose and echinate [31]. In addition, data of area, perimeter, aspect ratio, circularity, roundness and solidity in samples of the lateral and dorsal views and subsequent statistical analysis revealed differences between the three groups for all general morphological measurements [31]. The data obtained here for 51 new species of *Silene* classified them in the three mentioned groups plus a new group, named papillose. The four species classified as having papillose seeds (*S. holzmannii*, *S. laciniata*, *S. magellanica* and *S. perlmanii*) conformed a compact group well differentiated at a glance. They had distinctive morphological characteristics, such as the presence of long and broad prolongations (papillae) with mostly flat tips. The lowest values of circularity and solidity of the lateral and dorsal seed views for both individual and average seed silhouettes corresponded to this group as well.

Some differences were found between the results reported here and those given by Martín-Gómez et al. [31]. According to these authors, the lateral views of echinate seeds had the lowest values of circularity followed in an increasing order by smooth and rugose seeds. Our data showed that the echinate seeds also maintained the lowest circularity values among these three groups, but the highest values were obtained in the smooth seeds. This difference could be explained because the group of smooth seeds in this work has been reduced to four species, and one of them was convex (*S. baccifera*). In the previous report [31], although there was also a species characterized by convex seeds within the smooth seeds (*S. littorea*), this group was more numerous with a total of 12 species. Therefore, the lack of roughness of the outline of the seeds could be the main factor to support the high levels of circularity of the smooth seeds. Nevertheless, the two species *S. baccifera* and *S. littorea* would deserve further attention because they are the only species belonging to *S.* subg. *Behenantha* classified as smooth seeds. A high degree of homoplasy has been described in *Silene* [34]. This means a feature has been gained or lost independently in separate lineages over the course of evolution, compromising a correlation between phylogenetic analysis and specific morphological data. Several species from *S.* subg. *Behenantha* showed classification problems in phylogenetic analysis and morphological studies [29,34]. According to [34], specific morphological analysis have been recommended to complement *Silene* taxonomic work.

The main results of the measurements of aspect ratio agreed with the results published [31]. This similarity is especially related to the lateral seed views, since the seeds of smooth, rugose and echinate groups have showed similar data. However, the aspect ratio values of the dorsal views were less conserved, though there was a shared tendency with the highest values for the smooth seeds and the lowest, for echinate seeds. The analyses of roundness revealed a similar pattern after comparing with the previous data [31]. The values of the lateral views were quite similar, but not on the dorsal seed views, where despite the differences in the values obtained, the trend is entirely identical, with the highest data for the echinate seeds and the lowest ones for the smooth seeds. The observed differences of the specific values of the aspect ratio and roundness of the dorsal seed views could be based on the diversity of outline morphologies of these dorsal silhouettes. Nonetheless, the observed tendency was similar with independence of the studied species and samples, which would support the stability of the proposed seed classification based on the outline silhouettes. However, it would be recommendable to add more *Silene* samples to check the continuity of the observed tendencies, including samples for the new seed group, papillose.

Concerning solidity, the previous work [31] reported that the three groups of seeds had similar values for the lateral views, while the group of echinate seeds had higher values of solidity in the dorsal view. Our data pointed out that the solidity values of the lateral seed views were similar to the already obtained [31] for the smooth and rugose seeds (0.961 and 0.955 versus 0.958 and 0.957, respectively). However, the values were rather different for the echinate seeds (0.939 versus 0.955 [31]). This would be probably a consequence of the presence of larger spines of the seeds of the new studied species, and this possibility could be tested by the use of the average seed silhouettes. Regarding the dorsal seed views, the obtained values were only similar for the rugose seeds (0.943 versus 0.945 [31]), but the values of the smooth and echinate seeds were quite different (0.922 versus 0.894 [31] and 0.931 versus 0.955 [31], respectively). Despite these dissimilarities, the lowest values always corresponded to the smooth seeds, but the highest varied from echinate [31] to rugose. These changes and similarities would be explained by the presence or lack of concavities of the studied samples, since the solidity can reach the maximum value for convex figures [31]. Therefore, the absent or decreased of the presence of concavities were characteristic of convex seeds and hence the seeds with the maximum solidity values.

The analysis by the average seed silhouettes allowed to compare between morphological properties of seeds, as independent individuals, and those of a set of images grouped together. The use of average seed silhouettes reduced the morphological variation during the process of obtention of the data, which was remarkable for rugose and echinate seeds. Despite these differences, this comparison might reaffirm the morphological tendencies found in the analysis by groups of seeds based on individual seed silhouettes. For example, the papillose seeds showed the lowest values for circularity and solidity of the lateral and dorsal seed views for both individual and average seed silhouettes. In this work, we showed robust morphological tools for large sets of seeds. These tools resulted in similar clustering than phylogenetics classifications in *Silene* regarding subgenera and can be added to other tools developed for the analysis of *Silene* seeds [30,35]. The increasing number of species and populations of *Silene* will enable, eventually, a combination with phylogenetics for a better understanding of the *Silene* taxonomy.

Finally, two important aspects of this classification are: (1) the accurate description of the characters under analysis and (2) the stability of the characters under scrutiny. The overall shape may change with aging of seeds, and also a level of variation in the shape of the surface protuberances has been reported in *S. latifolia* subsp. *alba* [36]. According to this work, the shape of tubercles varied with the geographical location, and southern European populations tend to have lower and more rounded tubercles, while populations going towards Eastern Europe would have more conical tubercles and seeds with tall-conical, cylindrical, or tall-cylindrical tubercles characterize some collections from Hungary and Romania. Similarly, the section dedicated to *Silene* for Flora Iberica [37], reports obtuse tubercles for *S. latifolia* ([37], p. 396). We have worked so far with a total of 16 populations of *S. latifolia* from Central Europe and diverse locations in Southern Spain, France and Italy ([29,30,31,32] and unpublished results) and finding differences in the length and acuteness of the protuberances, but these seeds were all classified as echinate seeds based on the silhouette morphology. Although some individual seeds for some populations may lack echinate protuberances, in most of them the protuberances are obvious (pers. obs.). Further comparative population studies on seed morphology of well distributed species, as *S. latifolia*, would shed some light on the morphological variability of the seeds related to their geographical origin and even their ecological environments.

## 4. Materials and Methods

### 4.1. Seeds

A total of 51 species of *Silene* and one species of the close related genus *Eudianthe* were studied (see detailed data in Appendix D, Table A2). The seeds were obtained from the laboratories and botanical gardens indicated in Table A2 through a program of international cooperation with the Carpoespermateca of the Botanical Garden at the University of Valencia and were sent to IRNASA-CSIC in June, 2022. The plant nomenclature of the different taxonomical levels (subgenera, sections, species and subspecies) and their corresponding authorities were adapted according to Plants of the World Online (POWO) [2]. The taxonomical classification of the genus *Silene* followed Jaffari et al. [34].

### 4.2. Seed Images

Photographs of the lateral views of the seeds used in the images were taken with a Nikon Stereomicroscope Model SMZ1500 (Nikon, Tokio, Japan) equipped with a camera Nikon DS-Fi1 of 5.24 megapixels (Nikon, Tokio, Japan); lateral and dorsal views of these seeds used in the morphological analysis were taken with a camera Nikon Z6 equipped with an objective AF-S Micro NIKKOR 60 mm f/2.8G ED.

### 4.3. Seed Individual Outline and Average Silhouettes

The seed individual outlines and their corresponding average silhouettes for the lateral and dorsal views of the seeds in each species were obtained from 20 representative seed images by the method described [29,38]. A video describing the method used is available at: https://zenodo.org/record/4478344#.YzxbmExBxD8 (accessed on 1 December 2022).

### 4.4. General Morphological Description

The morphological characterization of the seed silhouettes of the different species of *Silene* followed the previous proposed classification of the outline of the seed silhouettes [31]. Based on this study, the *Silene* seeds were classified in three groups named: smooth, rugose, and echinate. The smooth seeds were characterized by the lack of superficial projections; the rugose seeds showed rounded projections or rounded tubercles; and those seeds with most of part of the silhouette highly projected with acute tubercles or spines were classified as echinate seeds.

Measurements corresponding to the lateral and dorsal views of seeds included: area (A), perimeter (P), length of the major axis (L), width (W), aspect ratio (AR is the ratio L/W), circularity (C), roundness (R) and solidity (S). They all were obtained with ImageJ [39]. A ruler was included in the photographs for the conversion of pixel units to length or surface units (mm or mm^2^). The measurements of length and width were omitted from the results because the data for area and aspect ratio gave enough information concerning seed size. Circularity index is the ratio (4πA)/P2 [40], while roundness is (4A)/πL2 [41]. Seed surface irregularities increase the perimeter and reduce the values of circularity, leaving roundness unaffected. On the other side, elongated seeds have decreased roundness values. These measurements were done for individual seed silhouettes and for average seed silhouettes, and hence, the total number of measured samples for each one was different. The comparison based on average silhouettes may reaffirm or refute the differences found in the analysis by groups of seeds. Area and perimeter values are excluded from this analysis because during the process of elaboration of the average silhouette, these aspects are submitted to random changes, while the shape of the average silhouette tends to represent a general trend in seeds.

### 4.5. Statistical Analysis

Mean, minimum and maximum and the standard deviation values were obtained for all the measurements indicated above (A, P, L, W, AR, C and R). Statistical analyses were done on IBM SPSS statistics v28 (SPSS 2021) and R software v4.1.2 [42]. As some of the data did not follow a normal distribution, non-parametric tests were applied for the comparison of populations. The Kruskal–Wallis test was used in the cases involving three or more groups, followed by stepwise stepdown comparisons by the ad hoc procedure developed by Campbell and Skillings [43]. P values inferior to 0.05 were considered significant. The coefficient of variation (CV) was calculated as CV = standard deviation/mean × 100 [44].

## 5. Conclusions

Based on optical photographs and the obtention of silhouettes from the seed images, the seeds of 51 *Silene* species have been classified in four well differentiated groups: smooth, rugose, echinate, and papillose. Morphological differences were found between the groups with circularity values in the lateral and dorsal seed views lower in the papillose and echinate seeds. The analysis involved the obtention of average silhouettes permitting to find those differences between groups that were more relevant. The differences found in circularity between the echinate and the smooth and rugose groups disappeared in the analysis with average silhouettes, while these between the papillose and the smooth and rugose groups were maintained after the analysis with average silhouettes. These morphological types for the lateral and dorsal views of the seeds may be related to the taxonomy of *Silene* based on DNA sequence analysis [31,34].

## Figures and Tables

**Figure 1 plants-11-03383-f001:**
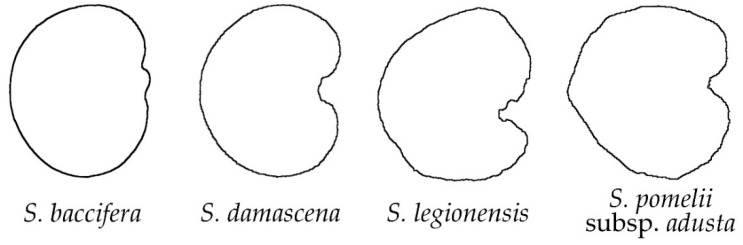
Silhouettes of the lateral views of seeds in the group of smooth outline.

**Figure 2 plants-11-03383-f002:**
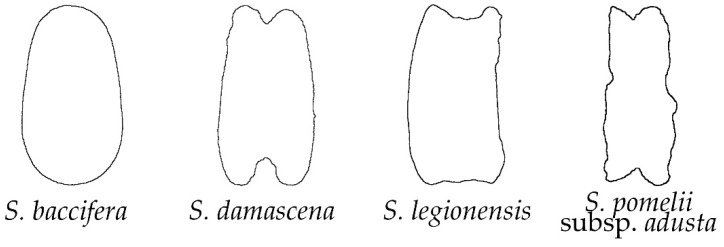
Silhouettes of the dorsal views of seeds in the group of smooth outline.

**Figure 3 plants-11-03383-f003:**
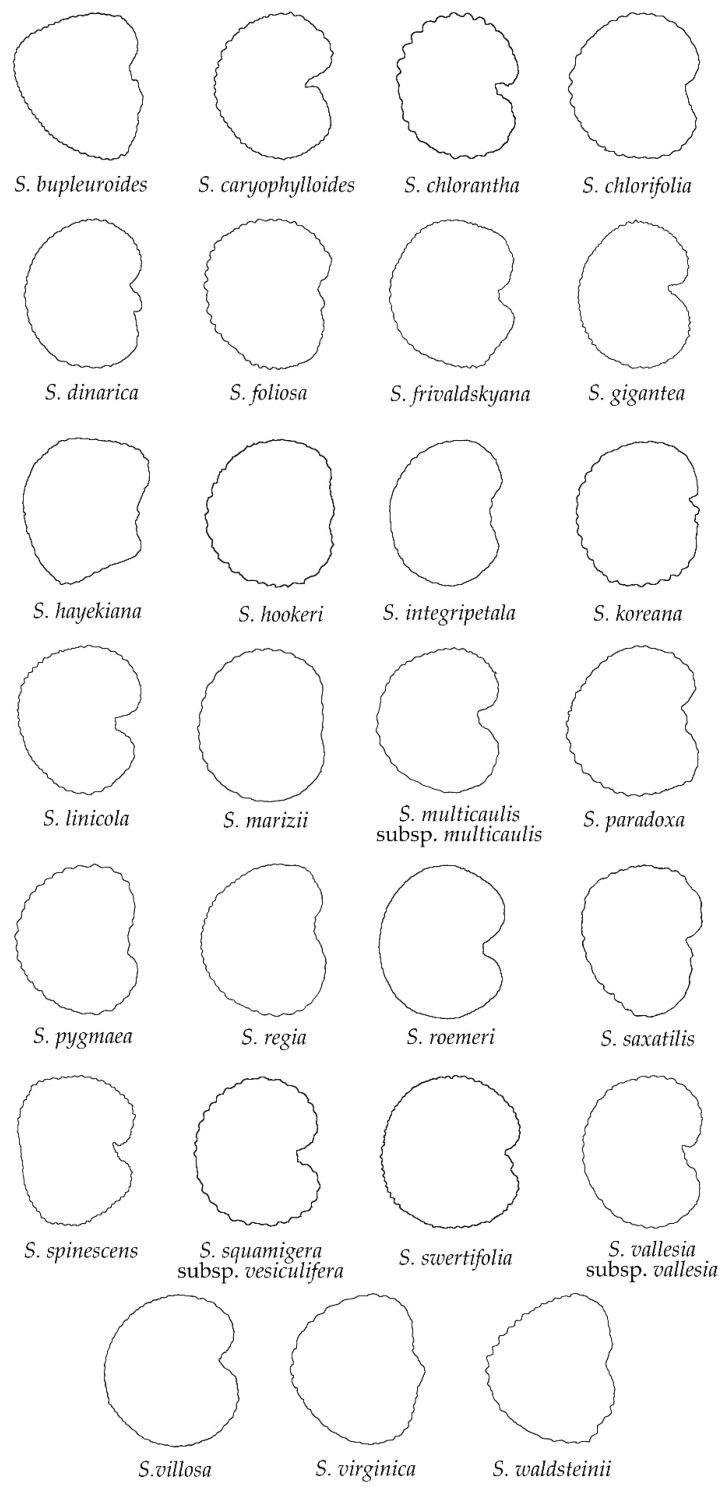
Silhouettes of lateral views of seeds in the group of rugose outline.

**Figure 4 plants-11-03383-f004:**
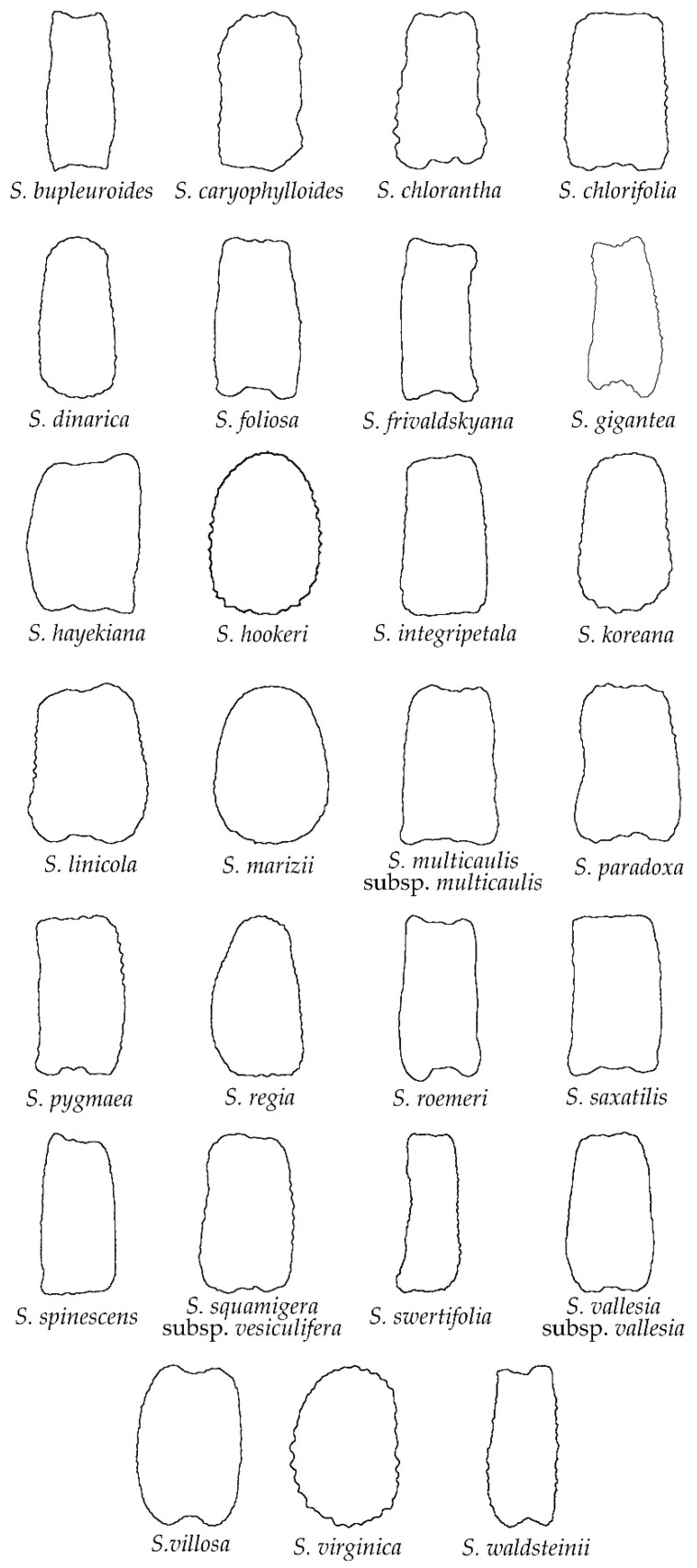
Silhouettes of the dorsal view of seeds in the group of rugose outline.

**Figure 5 plants-11-03383-f005:**
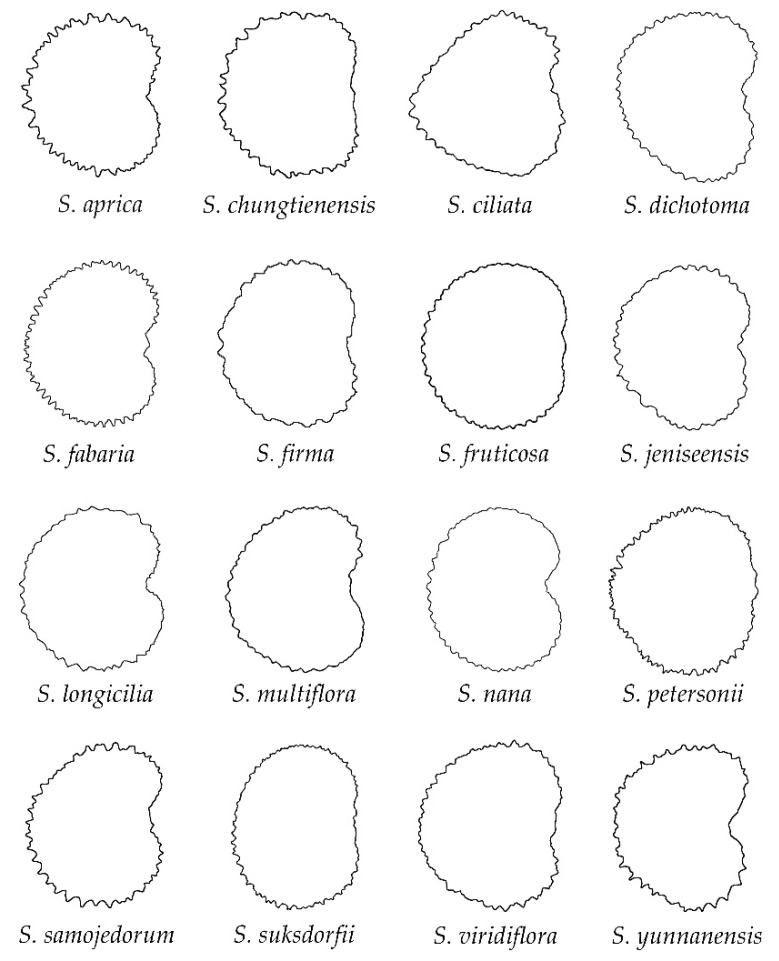
Silhouettes of the lateral view of seeds in the group of echinate outline.

**Figure 6 plants-11-03383-f006:**
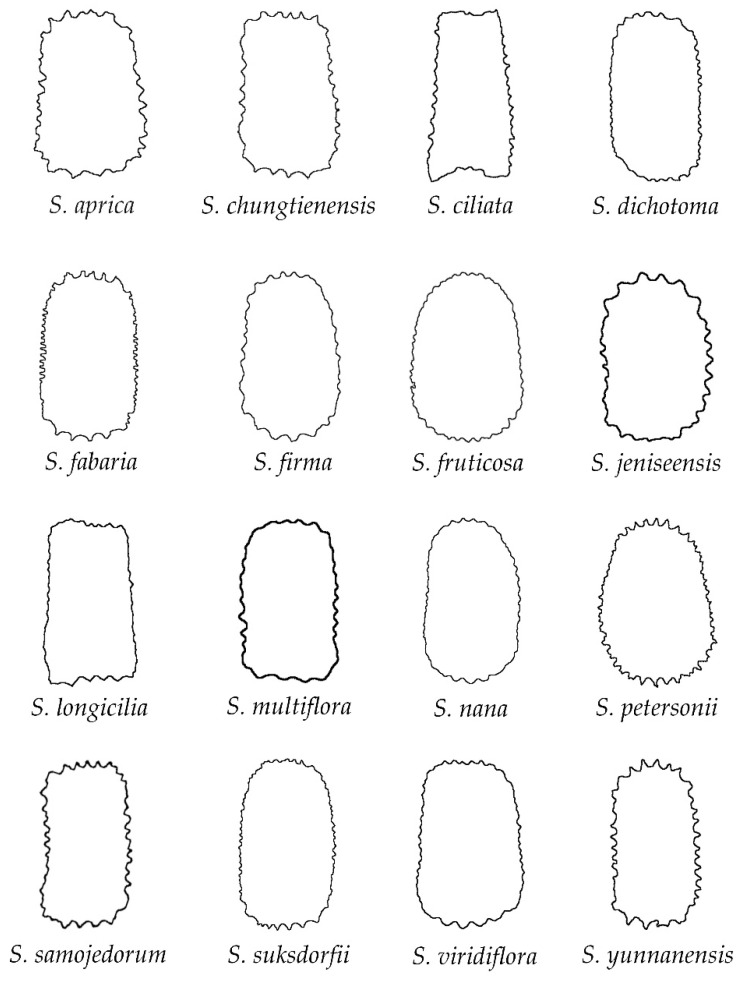
Silhouettes of the dorsal view of seeds in the group of echinate outline.

**Figure 7 plants-11-03383-f007:**
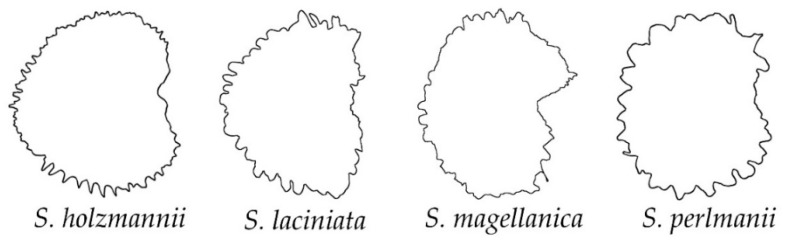
Silhouettes of the lateral view of seeds in the group of papillose outline.

**Figure 8 plants-11-03383-f008:**
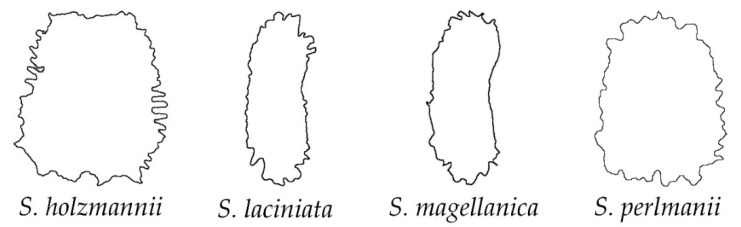
Silhouettes of the dorsal view of seeds in the group of papillose outline.

**Table 1 plants-11-03383-t001:** Summary of the values obtained for the different geometric parameters from the seeds of *Silene* for both dorsal and lateral seed views. *n* is the total number of seeds measured. SD = Standard Deviation. CV = Coefficient of Variation.

	Dorsal View (*n* = 1012)	Lateral View (*n* = 1025)
**Area**	Mean value: 0.83 mm^2^ (SD = 0.46; CV = 55.6) Min. value: 0.15 mm^2^ (*S. chungtienensis*) Max. value: 2.47 mm^2^ (*S. linicola*)	Mean value: 1.10 mm^2^ (SD = 0.63; CV = 57.2) Min. value: 0.21 mm^2^ (*S. chungtienensis*) Max. value: 4.28 mm^2^ (*S. swertiifolia*)
**Perimeter**	Mean value: 3.98 mm (SD = 1.18; CV = 29.6) Min. value: 1.72 mm (*S. chungtienensis*) Max. value: 8.62 mm (*S. fabaria*)	Mean value: 4.49mm (SD = 1.46; CV = 32.4) Min. value: 1.95 mm (*S. chungtienensis*) Max. value: 11.36 mm (*S. laciniata*)
**Circularity**	Mean value: 0.62 (SD = 0.11; CV = 17.1) Min. value: 0.25 (*S. holzmannii*) Max. value: 0.84 (*A. marizii*)	Mean value: 0.66 (SD = 0.13; CV = 19.1) Min. value: 0.21 (*S. holzmannii*) Max. value: 0.87 (*S. marizii*)
**Roundness**	Mean value: 0.57 (SD = 0.10; CV = 18.1) Min. value: 0.29 (*S. bupleuroides*) Max. value: 0.89 (*S. petersonii*)	Mean value: 0.81 (SD = 0.06; CV = 6.9) Min. value: 0.62 (*S. linicola*) Max. value: 0.98 (*S. paradoxa*)
**Aspect ratio**	Mean value: 1.80 (SD = 0.35; CV = 19.2) Min. value: 1.12 (*S. petersonii*) Max. value: 3.47 (*S. bupleuroides*)	Mean value: 1.25 (SD = 0.09; CV = 7.1) Min. value: 1.02 (*S. paradoxa*) Max. value: 1.63 (*S. linicola*)
**Solidity**	Mean value: 0.934 (SD = 0.03; CV = 3.6) Min. value: 0.76 (*S. bupleuroides*) Max. value: 0.991 (*S. baccifera*)	Mean value: 0.946 (SD = 0.02; CV = 2.3) Min. value: 0.805 (*S. perlmanii*) Max. value: 0.983 (*S. baccifera*)

**Table 2 plants-11-03383-t002:** Results of Kruskal–Wallis post hoc tests based on individual silhouettes for the comparison of the lateral views between the four morphological groups of seeds (smooth, rugose, echinate, and papillose) and *Eudianthe coeli-rosa*. Mean values and coefficient of variation (given in parentheses) are indicated for area (A), perimeter (P), circularity (C), and solidity (S). Values marked with the same superscript letter in each column correspond to populations that do not differ significantly at *p* < 0.05 (Campbell and Skillings test). N indicates the number of seeds analyzed.

Seed Type or Species	N	A	P	C	S
Smooth	73	1.31 ^b^ (45.2)	4.49 ^b^ (21.7)	0.77 ^d^ (8.2)	0.961 ^d^ (1.7)
Rugose	540	1.19 ^b^ (55.4)	4.47 ^b^ (28.3)	0.71 ^c^ (11.2)	0.955 ^c^ (1.3)
Echinate	323	0.89 ^a^ (59.4)	4.22 ^a^ (37.0)	0.60 ^b^ (17.1)	0.939 ^b^ (1.6)
Papillose	69	1.22 ^b^ (49.5)	6.24 ^c^ (27.1)	0.39 ^a^ (21.9)	0.892 ^a^ (2.2)
*E. coeli-rosa*	20	0.61 ^a^ (17.0)	3.48 ^a^ (10.2)	0.63 ^b^ (7.6)	0.936 ^b^ (0.8)

**Table 3 plants-11-03383-t003:** Results of Kruskal–Wallis post hoc tests based on individual silhouettes for the comparison of the dorsal seed views between the four morphological groups of seeds (smooth, rugose, echinate, and papillose), and *Eudianthe coeli-rosa*. Mean values and coefficient of variation (given in parentheses) are indicated for area (A), perimeter (P), aspect ratio (AR), circularity (C), roundness (R) and solidity (S). Values marked with the same superscript letter in each column correspond to populations that do not differ significantly at *p* < 0.05 (Campbell and Skillings test). N indicates the number of seeds analyzed.

Seed Type or Species	N	A	P	AR	C	R	S
Smooth	69	0.88 ^b^ (52.5)	4.01 ^b^ (19.6)	1.87 ^c^ (19.4)	0.64 ^c^ (20.2)	0.55 ^a^ (16.7)	0.922 ^b^ (5.6)
Rugose	540	0.90 ^b^ (52.9)	4.02 ^b^ (26.4)	1.86 ^c^ (18.3)	0.66 ^c^ (11.9)	0.56 ^a^ (17.8)	0.943 ^c^ (2.9)
Echinate	318	0.72 ^a^ (62.2)	3.73 ^a^ (34.9)	1.71 ^b^ (14.4)	0.59 ^b^ (14.9)	0.60 ^b^ (15.4)	0.931 ^b^ (3.1)
Papillose	65	0.83 ^b^ (33.2)	5.10 ^c^ (25.0)	1.82 ^a^ (33.4)	0.42 ^a^ (22.4)	0.61 ^c^ (27.6)	0.884 ^a^ (2.9)
*E. coeli-rosa*	20	0.45 ^a^ (7.5)	2.95 ^a^ (5.5)	1.60 ^a^ (5.8)	0.65 ^c^ (5.9)	0.63 ^c^ (5.6)	0.939 ^b,c^ (1.1)

**Table 4 plants-11-03383-t004:** Results of Kruskal–Wallis post hoc tests for the comparison between the average silhouettes of the lateral views in the four morphological seed groups (smooth, rugose, echinate, and papillose). Mean values and coefficient of variation (given in parentheses) are indicated for aspect ratio (AR), circularity (C), roundness (R) and solidity (S). Values marked with the same superscript letter in each column correspond to populations that do not differ significantly at *p* < 0.05 (Campbell and Skillings test). N indicates the number of average silhouettes analyzed.

Seed Type or Species	N	C	S
Smooth	4	0.76 ^b^ (5.6)	0.962 ^a,b^ (1.5)
Rugose	27	0.79 ^b^ (3.8)	0.971 ^b^ (0.8)
Echinate	16	0.77 ^b^ (7.0)	0.970 ^b^ (0.7)
Papillose	4	0.64 ^a^ (12.0)	0.946 ^a^ (1.6)

## Data Availability

Not applicable.

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
