# Peer review of "The Outline of Seed Silhouettes: A Morphological Approach to Silene (Caryophyllaceae)"

_plants, 2022, doi:10.3390/plants11233383_

Round 1

Reviewer 1 Report

Though, there is a free format submission, it was akward to read the article this way: Introduction, Results, Discussion, Material and Methods, and Conclusion.

If, authors agree, it would be better to organize the article according to the regular way: Introduction, Material and Methods, Results, Discussion and Conclusion. Knowing what was done, the appreciation of the Results obtained is much more better, as well as the Discussion and the Conclusion.

The information and photos shown in the Appendices section are quite good and give a good support for the information shown in the article. 

Author Response

Dear Reviewer,

Thank you very much for your commentaries and suggestions that have contributed to improve the quality of the article.

Concerning the presentation of the article and the order of the diverse sections, as you indicate, the article is organized in this way:

Introduction, Results, Discussion, Material and Methods, and Conclusion.

While we agree that in some cases it is desirable that the Materials and methods section is placed first, it is also true that placing the Results section first helps to expedite reading. Nevertheless, the organization of the manuscript was done following the template of the journal, and after consultation with the Editors, we must leave it in the present status, that corresponds to the standard procedure for this journal.

Looking forward to future collaboration. With best regards,

Emilio Cervantes

Reviewer 2 Report

In the manuscript entitled: "The outline of seed silhouettes: A new morphological approach to Silene (Caryophyllaceae)" (Number: plants-2072636), the Authors present the classification of seeds of the 51 species from genus Silene based on morphological features (surface relief) and measurements of the lateral seed views. The title of the work states that it will be presented: "A new morphological approach...", but in my opinion, the authors propose a new morphological form of the seed, namely the papillose seeds not “new approach”. Please verify the title.

 The Results section:

Line 141: “Echinate seeds are represented by the following 16 species (Figure 5)” – unclear sentence. How are seeds represented by species?

Line 144: “Behenantha is represented by 10 species and includes most of the species in this group (10 out of 16, 62.5%),” – unclear, what kind of “group”?

 The study and description of the morphology of the seed surface relief should be based on analyses conducted by scanning electron microscopy (SEM), especially since the authors propose the introduction of a new form - the papillose seeds. Unfortunately, the paper lacks such analyses therefore it is difficult to verify the authors' reports. Thus, the publication should be extended to SEM analysis, and I strongly recommend add new analyses.

 The Materials and methods section:

There is no information in what condition the seeds were analyzed (whether fresh - just after harvesting, or dry, and under what conditions they were stored). 

             In conclusion, in my opinion, the description of seeds morphology is an important source of information for plant taxonomy. However, the manuscript contains several deficiencies that need to be completed, clarified and new data should be provided.

Author Response

Dear Reviewer,

Thank you very much for your commentaries and suggestions that have contributed to improve the quality of the article.

Two major criticisms of the evaluation report are: 1) The title: "A new morphological approach...", and 2) The indication that the study and description of the morphology of the seed surface relief should be based on analyses conducted by scanning electron microscopy (SEM).

Concerning the first point, and following your advice, the adjective “new” has been deleted from the title. In relation with the second, we need to argue in favor of our method.

While it is true that most of the analyses reported so far of seed surface morphology in Silene have been done by scanning electron microscopy, it is also true that an approach to morphological analysis can also be done based on optical photographs, and this is one of the major innovations of this work. With this basis, the results here presented can be well reproduced and verified.

In addition, the method here presented, based on optical photographs, has several advantages: It is easy, requiring only very basic and accessible equipment, and in consequence it can be applied to higher number of samples and by independent botanical researchers or small laboratories that are not equipped with electron microscopy, but can have more easily access to photographic cameras.

In summary, this method may serve as a stimulus to the morphological analysis and applications in groups where no SEM is available or to analyze larger number of samples in a preliminary analysis before doing more detailed work on SEM.

The other suggested changes have been done as follows:

The sentence:

“Echinate seeds are represented by the following 16 species (Figure 5)” – unclear sentence.

Has been changed to:

“The group of echinate seeds is integrated by the following 16 species (Figure 5)”

The following sentence:

“S. subg. Behenantha is represented by 10 species and includes most of the species in this group (10 out of 16, 62.5%), while S. subg. Silene is represented by six species.”

Has been changed to:

“Most of the species in this group belong to S. subg. Behenantha (10 species out of 16, 62.5%),..”

Concerning this important question:

“There is no information in what condition the seeds were analyzed (whether fresh - just after harvesting, or dry, and under what conditions they were stored).” 

In Materials and Methods, 4.1. Seeds, line 375, The following sentence:

“The seeds used were obtained from the seed collection kept at the Botanical Garden of the University of Valencia (Spain).”

Has been changed to:

“The seeds were obtained from the laboratories and botanical gardens indicated in Table C2 through a program of international cooperation with the Carpoespermateca of the Botanical Garden at the University of Valencia, and were sent to IRNASA-CSIC in June, 2022.”

Thus, the seeds were stored in different conditions in their diverse laboratories of origin.

Thanking you very much for your attention and interest, yours sincerely,

Emilio Cervantes
